genetics

microhaplotype, genetic markers, individual identification

**Author for correspondence:**
Lagabaiyila Zha
e-mail: 40409716@qq.com

†These authors contributed equally to this work and should be considered as co-first authors.

# Construction and forensic application of 20 highly polymorphic microhaplotypes

Aliye Kureshi[1,†], Jienan Li[3,†], Dan Wen[3], Shule Sun[3], Zedeng Yang[3] and Lagabaiyila Zha[2,3,1]

[1]School of Basic Medical Sciences, Xinjiang Medical University, Urumqi 830011, Xinjiang, People's Republic of China
[2]Shanghai Key Laboratory of Forensic Medicine, Academy of Forensic Science, Shanghai 200063, People's Republic of China
[3]Department of Forensic Medicine, School of Basic Medical Sciences, Central South University, No. 172, Tongzipo Road, Changsha 410013, Hunan, People's Republic of China

AK, 0000-0001-6929-3372; JL, 0000-0003-4485-1707; LZ, 0000-0003-2207-4441

Microhaplotype markers have become an important research focus in forensic genetics. However, many reported microhaplotype markers have limited polymorphisms. In this study, we developed a set of highly polymorphic microhaplotype markers based on tri-allelic single-nucleotide polymorphisms. Eleven newly discovered microhaplotypes along with nine previously identified in our laboratory were studied. The microhaplotype genotypes of unrelated individuals and familial samples were generated on the MiSeq PE300 platform. These 20 loci have an average greater than 3.5 effective number of alleles. Over the whole set, the cumulative power of discrimination was $1–3.3 \times 10^{-18}$, the cumulative power of exclusion was $1–1.928 \times 10^{-7}$ and the theoretical probability of detecting a mixture was $1–1.427 \times 10^{-6}$. Differentiation comparisons of 26 populations from the 1000 Genomes Project distinguished among East Asian, South Asian, African and European populations. Overall, these markers enrich the current microhaplotype marker databases and can be applied for individual identification, paternity testing and biogeographic ancestry distinction.

## 1. Introduction

Single-nucleotide polymorphisms (SNPs) are the most abundant variations in the human genome [1]. There are millions of SNPs in each individual, making them significant in forensic research, especially for the identification of individuals [2]. They have many useful features. First, the amplicons of SNPs are smaller than commonly used short tandem repeats (STRs), and this may be helpful when analysing degraded samples. Second, SNPs tend to be specific to certain populations, making them

promising genetic markers for inferring ancestry. Moreover, their low mutation rates [3] make them useful in paternity testing [4]. However, SNPs are mainly biallelic markers with limited polymorphic content [5,6]. To establish a new forensic marker that expresses more polymorphism than single SNPs, Pakstis *et al*. [7] proposed a multi-SNP haplotype system called mini-haplotype. This is defined as three or more SNPs with high heterozygosity within a molecular region less than 10 kb. However, the segment size of the mini-haplotype is too large for detection in forensic laboratories. On the basis of mini-haplotype, Kidd *et al*. [8] optimized the concept of the microhaplotype to fit the application of forensic science. A microhaplotype locus is a short segment of DNA (smaller than 200 bp) composed of two or more SNPs that produces a multi-allelic haplotype [8]. Recombination rates among SNPs are quite low in such a short region, and massively parallel sequencing (MPS) can be used to identify phase-known haplotypes in a single sequence run [9]. Microhaplotype loci with improved polymorphisms and low mutation rates are being widely studied for their potential use to supplement the use of traditional forensic genetic markers [10–13].

Nonetheless, at present, STRs are the preferred markers used in forensic genetics owing to their multi-allelic nature and thus high number of polymorphisms [14]. Capillary electrophoresis (CE) is generally used for detection when applying STR genotyping in forensic genetics. However, STRs have high mutation rates, and are not ideal for ancestry identification [15,16]. Their mutation rates are $10^3–10^4$ times those of SNPs [17], which lead to false exclusion in paternity testing [18]. STRs often generate artificial peaks such as stutter peaks and -A peaks in CE analyses, which may affect the analysis of unbalanced DNA mixtures [19]. STR detection through MPS technology has disadvantages such as read length limitations of most MPS platform, homopolymer sequencing errors generated during STR sequencing and complex data interpretation [20–22]. There are no such problems with microhaplotypes [23–25]. Therefore, microhaplotypes could be great supplementary tools for STRs in forensic science.

A number of microhaplotypes have been proposed [25–28], but many have a limited number of polymorphisms. In this study, we constructed highly polymorphic microhaplotypes consisting of tri-allelic SNPs. Then we explored their applicability in terms of identifying individuals, determining biological relationships and detecting DNA mixtures using the MiSeq PE300 platform (Illumina, San Diego, CA, USA). We also used them to infer biogeographic ancestry based on 1000 Genomes Phase 3 data [29]. The populations considered were from five main regions: East Asian (EAS), including Han Chinese in Beijing (CHB), Han Chinese in Southern China (CHS), Chinese Dai in Xishuangbanna (CDX), Kinh in Ho Chi Minh City Vietnam (KHV) and Japanese in Tokyo, Japan (JPT); African (AFR), including African Caribbeans in Barbados (ACB), Americans of African ancestry in southwestern USA (ASW), Esan in Nigeria (ESN), Gambian in Western Divisions in Gambia (GWD), Mende in Sierra Leone (MSL), Luhya in Webuye, Kenya (LWK) and Yoruba in Ibadan, Nigeria (YRI); South Asian (SAS), including Bengali from Bangladesh (BEB), Gujarati Indian from Houston, TX, USA (GIH), Indian Telugu from the UK (ITU), Sri Lankan Tamil from the UK (STU) and Punjabi from Lahore, Pakistan (PJL); European (EUR), including residents of Utah, USA with Northern and Western European Ancestry (CEU), British in England and Scotland (GBR), Finnish in Finland (FIN), Iberian population in Spain (IBS) and Toscani in Italy (TSI); and American (AMR), including Colombians from Medellín, Colombia (CLM), Mexican ancestry from Los Angeles, USA (MXL), Puerto Ricans from Puerto Rico (PUR) and Peruvians from Lima, Peru (PEL).

# 2. Material and methods

## 2.1. Candidate loci selection and primer design

SNPs, with a preference for tri-allelic ones, were selected according to the following criteria: (i) for Chinese Han populations (CHB and CHS from the 1000 Genomes Project), a minor allele frequency (MAF) greater than 0.10, and (ii) SNPs on the same microhaplotypes with an identical allele frequency were excluded. Then each microhaplotype needed to be less than 200 bp, with a molecular distance between loci on the same chromosome greater than 2.0 Mb to minimize the effects of linkage disequilibrium. The effective number of alleles ($A_e$) needed to be greater than 3.0; and heterozygosity for each microhaplotype less than or equal to 0.6. The naming of these microhaplotypes followed the principles proposed by Kidd [30]; those in the same molecular region with different SNP compositions were distinguished from each other using lower-case letters (a, b, c, …). The specific amplification primers were designed using Primer Premier5.0 and Oligo software v. 2.3.7 (Molecular Biology Insights, Colorado Springs, CO, USA). Finally, BLAST was used to verify amplicons homology.

## 2.2. DNA samples

Blood samples were collected from Chinese Han populations who provided written informed consent (ethics approval code: 2018-S194), including 50 unrelated individuals and 12 parent/child duos.

## 2.3. MPS and data analysis

All samples were amplified in a SmartChip using the Takara/WaferGen SmartChip TETM system (Takara Bio, Kusatsu, Japan). Parallel nanolitre polymerase chain reaction (PCR)-based target enrichment for amplicon sequencing was performed using a method similar to that described in De Wilde *et al.* [31]. The PCR system (100 nl per well) for each sample comprised MasterMix 1×, Universal Outer Primer 1 µM, Index Primer 1 µM, Inner Primer Pair 0.25 µM and DNA template 2.5 ng µl$^{-1}$. PCR was performed on a T100 Thermal Cycler (Bio-Rad Laboratories, Hercules, CA, USA) with the following conditions: 95°C for 5 min, 10 cycles at 95°C for 15 s, 60°C for 30 s, 72°C for 60 s, 2 cycles at 95°C for 15 s, 80°C for 30 s, 60°C for 30 s, 72°C for 60 s, 8 cycles at 95°C for 15 s, 60°C for 30 s, 72°C for 60 s, 2 cycles at 95°C for 15 s, 80°C for 30 s, 60°C for 30 s, 72°C for 60 s, 8 cycles at 95°C for 15 s, 60°C for 30 s, 72°C for 60 s, 10 cycles at 95°C for 15 s, 80°C for 30 s, 60°C for 30 s and 72°C for 60 s. The PCR products were purified by gel-cut recovery. All samples were sequenced on the MiSeq PE300 platform according to the manufacturer's recommendations.

The base coverage threshold of sequencing was set to 30×. The raw data were processed with bcl2fastq software for each sample and run through the BBDuk software of BBMap v. 37.75 (https://sourceforge.net/projects/bbmap). The phase-known genotype data were ascertained using GATK v. 4.0 [32] and HapCUT2 [33]. To verify the reproducibility of sequencing results, 30 samples were re-sequenced on another chip.

## 2.4. STR genotyping

The DNA samples were amplified using a Goldeneye 20A kit (Peoplespot, Beijing, China) with a 9700 Thermal Cycler (Thermo Fisher Scientific, Waltham, MA, USA). PCR products were separated and detected using an ABI PRISM 3130xl Genetic Analyzer (Applied Biosystems, Foster City, CA, USA). The genotypes were analysed using GeneMapperID v. 3.2 (Applied Biosystems).

## 2.5. Sanger sequencing

Sequencing accuracy was validated through T vector molecular cloning and Sanger sequencing. Ten randomly selected loci were typed using the S14 sample and checked for consistency against the sequencing result of the S14 sample using the MiSeq PE300 platform.

## 2.6. Statistical analysis

The forensic parameters were evaluated using modified Powerstats software v. 1.2 [34] based on the sequencing results of 50 unrelated individuals, including the power of discrimination (PD), power of exclusion (PE), observed heterozygosity (Ho) and *p*-value of exact tests for Hardy–Weinberg equilibrium (HWE). Kidd & Speed [35] defined the effective number of alleles ($A_e$) for a locus as the equivalent number of neutral alleles of equal frequency, calculated using the formula $1/\sum p_i^2$ (where $p_i$ represents the frequency of allele i). The probability of detecting DNA mixtures was calculated as well. Linkage disequilibrium (LD) between loci was estimated with $\chi^2$-tests using Arlequin v. 3.5 software [36], and correlation coefficients ($r^2$) for loci pairs were calculated using the SHEsis online tool [37]. SNP information on 26 populations from 1000 Genomes Phase 3 data was used for estimating haplotypes and haplotype frequencies with PHASE v. 2.1.1 [38,39]. We also calculated the principal forensic parameters for all 26 populations to assess the applicability of the set of microhaplotype markers to different populations. STRUCTURE software v. 2.3.4 [40] was used to evaluate their utility for inferring ancestry. The program was run three times with 10 000 burn-ins and 50 000 Markov chain Monte Carlo iterations for each *K* value (*K* = 2–7); CLUMPP v. 1.1.2 [41] and Distruct v. 1.1 [42] were used to visualize the results. We applied the neighbour-joining (NJ) method [43] to establish a phylogenetic tree using POPTREE2 [44] and MEGA v. 7.0 [45]. *F*-statistic values were calculated using Arlequin v. 3.5 software, and R software v. 3.4.2 was used to describe *F*-st among populations with the 'pheatmap' package. For the 12 parent/child duos, paternity was

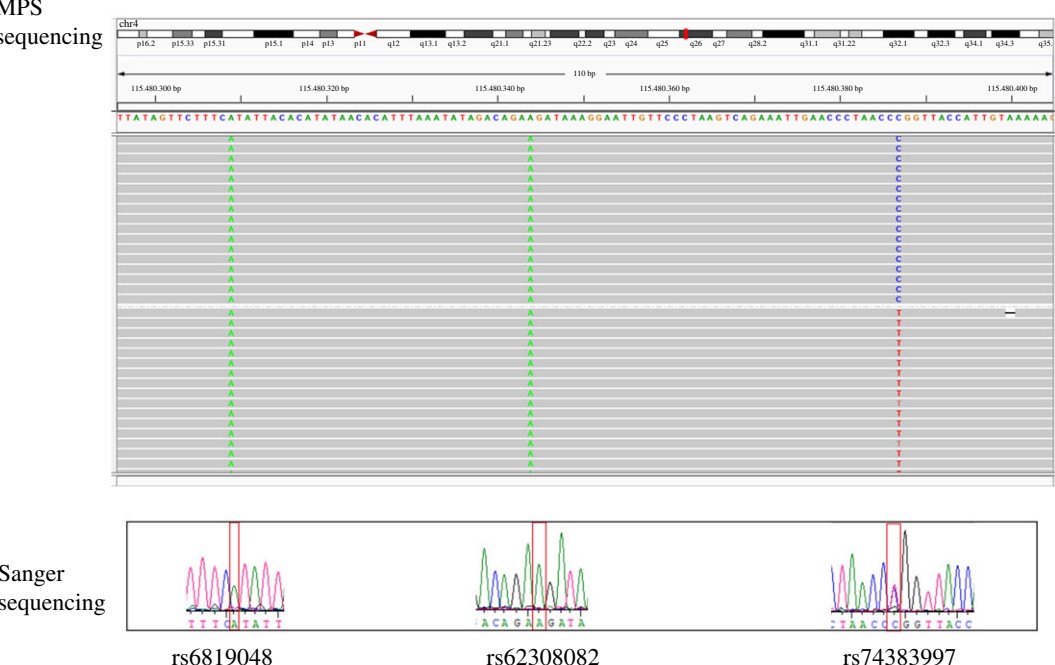

**MPS sequencing**

**Sanger sequencing**

rs6819048          rs62308082          rs74383997

**Figure 1.** The MPS sequencing result and Sanger sequencing result of the S14 sample at mh04zha007 site.

separately verified using the combined paternity index (CPI) based on STR genotyping and MiSeq sequencing results.

# 3. Results and discussion

## 3.1. Marker selection and evaluation

After excluding loci according to the screening criteria and sequencing quality control threshold, 20 microhaplotypes were successfully sequenced on the MiSeq PE 300 platform. The accuracy of MPS sequencing was verified on the S14 sample for 10 randomly selected loci. The results are presented in figure 1.

The newly proposed markers identified in this study are mh02zha012, mh04zha001, mh04zha002, mh04zha007, mh08zha011, mh09zha008, mh11zha006a, mh10zha002, mh14zha003, mh17zha001 and mh22zha008. Table 1 lists the basic information and forensic parameters of the 20 microhaplotypes. All loci consisted of three or more SNPs with one tri-allelic SNP, except for locus mh22zha008. The molecular lengths of the 20 loci ranged from 8 to 178 bp; 13 that were less than 150 bp might be useful for slightly degraded DNA samples, especially mh14zha003 which was only 8 bp. The detailed information of specific primers and PCR amplicon sizes are reported in electronic supplementary material, table S1.

The HWE and LD test results are given in electronic supplementary material, table S2. There was no significant deviation from HWE after Bonferroni correction ($p = 0.05/20 = 0.0025$). The LD $p$-values of microhaplotype markers on the same chromosome showed no significant deviation from expectations, suggesting that these sites were in linkage equilibrium. To further evaluate LD, we calculated another parameter, $r^2$ (electronic supplementary material, figure S1). The $r^2$ values between marker pairs on the same chromosome were all under 0.04, supporting the previous conclusion of LD tests.

The $A_e$ values of the 20 microhaplotypes ranged from 2.818 (mh04zha001) to 4.995 (mh19zha007), with an average value of 3.724, suggesting wide applicability of this system in forensic practice [35]. We compared the average $A_e$ value and the matching probability (MP, the probability that two randomly selected individuals have the same genotype at the tested locus) of the set with other microhaplotypes proposed in table 2. $A_e$ values correlate with the ability of microhaplotype loci to detect and deconvolute DNA mixtures [46]. For instance, when a microhaplotype locus with an $A_e$ value of 3.0 is applied for detecting a mixture of two unrelated individuals, the probability of there being a third allele was 0.4444 under the simple HWE model [35]. Hence, the maximum probability of detecting a mixture for this locus was 0.4444; for a locus with an $A_e$ value of 4.0, the maximum probability would be 0.65625. We used the minimal integral value of $A_e$ for our probability calculation. The cumulative probability of detecting a mixture with the set of

**Table 1.** Detailed information and forensic parameters of 20 microhaplotypes. Microhaplotypes (MHs); Chr (chromosome of microhaplotypes); position (nucleotide position of microhaplotypes based on build 37); SNPs (SNPs ID); Len (molecular length); effective number of alleles ($A_e$); power of discrimination (PD); power of exclusion (PE); and observed heterozygosity (Ho).

| MHs | Chr | position | Len | SNPs | $A_e$ | Ho | PD | PE |
|---|---|---|---|---|---|---|---|---|
| mh02zha012 | 2 | 146 369 062–146 369 166 | 105 | **rs949778**/rs867005/rs952210/ | 3.021 | 0.70 | 0.801 | 0.428 |
| mh03zha001 | 3 | 25 069 341–25 069 517 | 177 | rs4858685/**rs4858686**/rs75773180/rs9838878/ | 3.247 | 0.72 | 0.838 | 0.460 |
| mh04zha001 | 4 | 9 589 873–9 589 956 | 84 | **rs6830692**/rs9714725/rs12501341/rs10939388/ | 2.818 | 0.70 | 0.803 | 0.428 |
| mh04zha002 | 4 | 14 837 753–14 837 841 | 89 | rs10939597/rs9276692/**rs62409414**/rs62409415/ | 4.647 | 0.84 | 0.858 | 0.675 |
| mh04zha004 | 4 | 57 939 863–57 940 018 | 156 | rs1004992/**rs1914740**/rs1714017/rs6835177/ | 3.709 | 0.68 | 0.887 | 0.398 |
| mh04zha007 | 4 | 115 480 309–115 480 387 | 79 | **rs6819048**/rs62308082/rs74383997/ | 3.484 | 0.74 | 0.854 | 0.493 |
| mh05zha004 | 5 | 174 968 560–174 968 732 | 173 | rs2457087/rs2644662/**rs2662178**/ | 3.251 | 0.70 | 0.846 | 0.428 |
| mh07zha003 | 7 | 41 441 508–41 441 607 | 100 | rs4724041/rs378367/**rs433709**/rs404569/ | 4.129 | 0.76 | 0.902 | 0.527 |
| mh07zha004 | 7 | 44 191 190–44 191 346 | 157 | **rs6971410**/rs2971679/rs3808323/ | 3.971 | 0.76 | 0.889 | 0.527 |
| mh07zha009 | 7 | 103 851 964–103 852 139 | 176 | rs144858626/rs149890778/rs11773043/**rs7792859**/ | 3.133 | 0.64 | 0.856 | 0.342 |
| mh08zha011 | 8 | 13 728 306–13 728 400 | 95 | rs4831247/rs13265601/**rs4831248**/rs13268053 | 4.219 | 0.86 | 0.887 | 0.715 |
| mh09zha008 | 9 | 115 934 698–115 934 852 | 155 | **rs11506774**/rs1098166/rs10739387/ | 3.820 | 0.66 | 0.878 | 0.369 |
| mh10zha002 | 10 | 20 178 703–20 178 809 | 107 | rs10764175/rs148665640/rs10827896/**rs10827897** | 3.903 | 0.82 | 0.862 | 0.637 |
| mh11zha006a | 11 | 124 823 941–124 824 066 | 126 | rs3809057/**rs3809056**/rs3809055/rs3809054/ | 3.834 | 0.76 | 0.876 | 0.527 |
| mh14zha003 | 14 | 72 252 135–72 252 142 | 8 | rs4902946/rs8012670/**rs4902947** | 3.674 | 0.70 | 0.886 | 0.428 |
| mh16zha009 | 16 | 86 921 457–86 921 568 | 112 | rs76047588/**rs11641186**/rs11641193/rs80213582/ | 3.519 | 0.88 | 0.876 | 0.755 |
| mh17zha001 | 17 | 239 921–240 040 | 120 | rs5602344/**rs4131415**/rs4260117 | 3.808 | 0.64 | 0.874 | 0.562 |
| mh19zha007 | 19 | 28 888 223–28 888 363 | 141 | **rs8106726**/rs8102417/rs59490836/rs10406130/ | 4.995 | 0.82 | 0.916 | 0.637 |
| mh19zha009 | 19 | 53 632 326–53 632 503 | 178 | rs74178308/rs8108729/**rs8107824**/rs8108835/rs2560950/ | 3.737 | 0.80 | 0.867 | 0.599 |
| mh22zha008 | 22 | 50 502 766–50 502 888 | 123 | rs11568183/rs8142282/rs8136173/ | 3.559 | 0.76 | 0.858 | 0.527 |

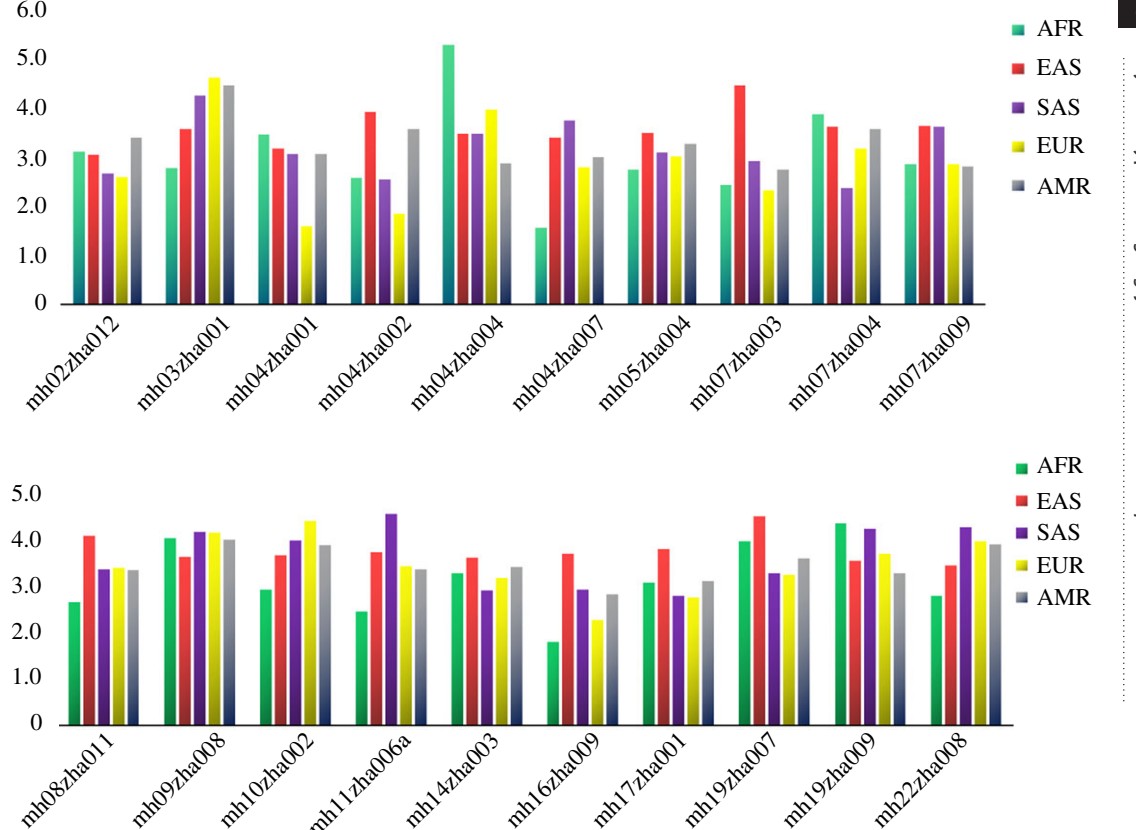

**Figure 2.** The average value of $A_e$ for five main continents from 1000 Genomes Project.

**Table 2.** The comparison of forensic parameters between microhaplotype markers.

| number of loci | populations | average value of $A_e$ | average value of MP | reference |
|---|---|---|---|---|
| 87 | 100 Italians | 3.043 | 0.2396 | Turchi *et al.* [27] |
| 25 | 60 unrelated Chinese Han individuals | 3.230 | 0.1622 | Chen *et al.* [11] |
| 26 | CHB populations from 1000 Genomes | 3.571 | 0.1387 | Chen *et al.* [28] |
| | CHS populations from 1000 Genomes | 3.566 | 0.1411 | |
| 20 | CHB populations from 1000 Genomes | 3.762 | 0.1228 | in this study |
| | CHS populations from 1000 Genomes | 3.721 | 0.1281 | |
| | 50 unrelated Chinese Han individuals | 3.724 | 0.1342 | |

microhaplotypes was equal to $1 - (1 - 0.4444)^{15} (1 - 0.65625)^4 = 0.999997927$. $A_e$ values convey information on the polymorphisms of markers, and we assessed how these could be used to analyse DNA mixtures. The larger the $A_e$ values reflect the better capacity of detecting mixed DNA samples. The PD of 20 loci ranged from 0.801 to 0.916 with an average value of 0.866. The cumulative PD value of the set was $1-3.3 \times 10^{-18}$. The PE values of those loci ranged from 0.342 to 0.755 with a mean value of 0.523, and the cumulative PE was $1-1.928 \times 10^{-7}$. The Ho values of all loci were greater than 0.6. The observed alleles of microhaplotypes and allele frequencies are illustrated in electronic supplementary material, figure S2; most microhaplotypes had at least four alleles, although the maximum number was 12 for locus mh04zha004.

The principal forensic statistics are summarized in electronic supplementary material, table S3. The combined MP of 26 populations was calculated following the method proposed by Balding & Nichols [47], and ranged from $9.57 \times 10^{-4}$ (MSL population) to $1.04 \times 10^{-12}$ (STU population). The combined MP of CHB population for unrelated individuals was $8.73 \times 10^{-11}$, suggesting this set can be used independently for personal identification. Alleles observed in the global populations and allele

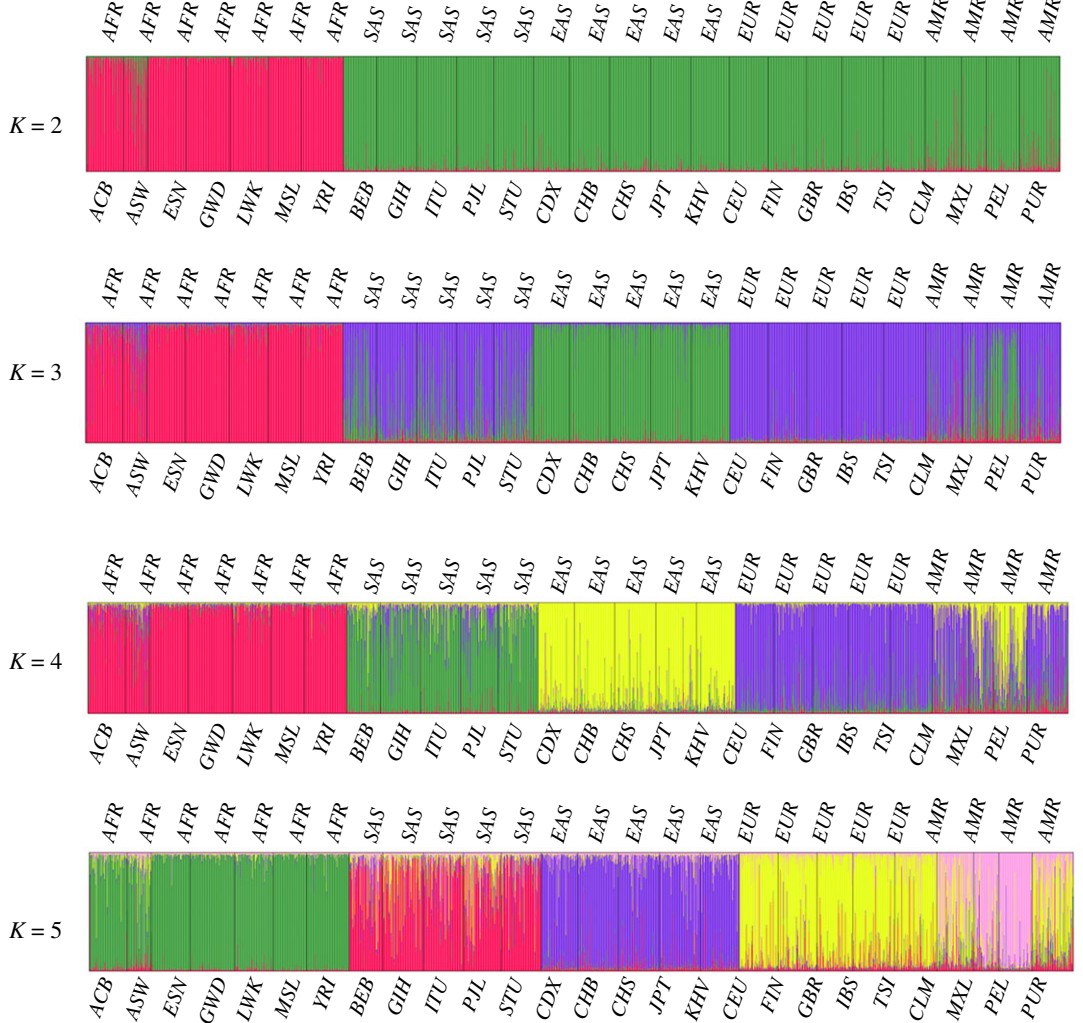

**Figure 3.** The STRUCTURE analysis of 26 populations based on the set of microhaplotypes.

frequencies are presented in electronic supplementary material, table S4, the allele frequencies of tri-allelic SNPs for every microhaplotype are given in electronic supplementary material, table S5. We speculate that the tri-allelic SNPs in microhaplotypes contribute significantly to the polymorphism of each locus in the Chinese Han population. The $A_e$ values of 20 loci were calculated for 26 populations, and average $A_e$ for five main regions are depicted in figure 2. Note that EAS populations all have an average $A_e > 3.0$ at all loci. More population genetic studies of this highly polymorphic panel will be done in the future, so that this panel could be applied in forensic casework.

## 3.2. Biogeographic ancestry distinction

The results of STRUCTURE analysis are shown in figure 3. At $K = 2$, the AFR populations (ACB, ASW, MSL, GWD, LWK, ESN and YRI) were distinguished from the others. At $K = 3$, it was possible to find genetic differences between AFR and EAS. At $K = 4$, the four populations of AFR, SAS (BEB, GIH, ITU, STU and PJL), EAS and EUR (GBR, FIN, CEU, IBS and TSI) were separated, but AMR (PEL, MXL, CLM and PUR) populations were not separated from EUR. At $K = 5$, populations of AMR and EUR formed two mixed clusters that could be attributed to the immigration history of the AMR population from Europe. Another reason for poor differentiation might be the small number of loci and the deficiency of markers' ancestry information. Because the set of microhaplotypes was not specifically designed for inferring ancestry, we focused more on $A_e$ values than Rosenberg's informativeness ($I_n$) values [46]. The heatmap of $F$-st is illustrated in electronic supplementary material, figure S3. The AFR populations clustered in the upper left part of the figure with negligible $F$-st values. Conversely, there was a high $F$-st value between AFR and EAS populations, representing significant genetic differentiation

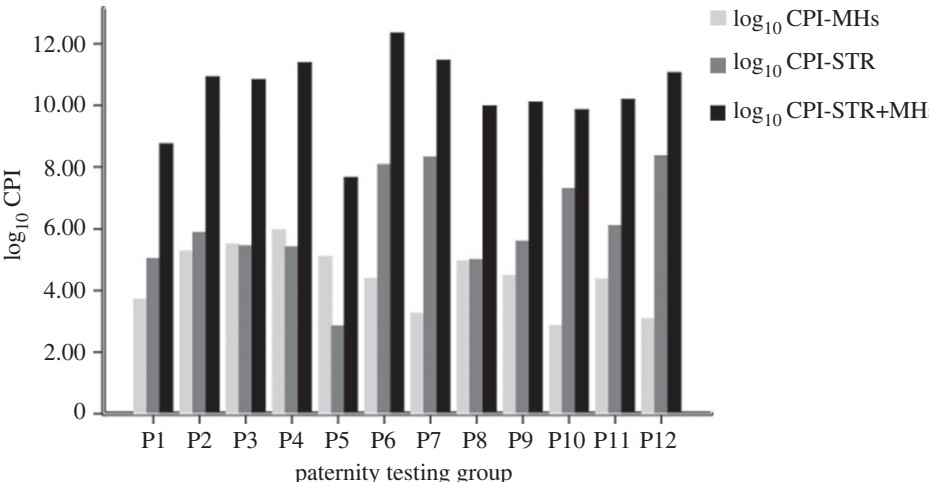

**Figure 4.** Comparison of $\log_{10}$ value of CPI.

among African and East Asia populations. A phylogenetic tree was constructed using the NJ method (electronic supplementary material, figure S5); it produced five main branches (basically consistent with geographical distribution) extending from a rooted tree starting with AFR populations. Taken together, these results indicate that our system unambiguously differentiated between four major populations: East Asian, African, South Asian and European/American.

## 3.3. Determination of biological relationships

The specific genotypes and CPI values of 12 parent/child duos based on microhaplotype sequencing and CE of STR markers are shown in electronic supplementary material, table S6. The genotypes of 20 microhaplotype loci for all duos are in accordance with Mendel's law of inheritance. The CPI value of eight duos (P2, P3, P4, P5, P6, P8, P9 and P11) exceeded the threshold value of 10 000, which could be direct confirmation of paternity. Furthermore, we compared the $\log_{10}$ values of CPI using a single marker type (microhaplotype or STR) with those using STR markers with our set of microhaplotypes and show the results in figure 4 (TPOX loci were ruled out from final cumulative operation based on LD test results). The combined CPI values all exceeded 10 000. For group P5, the CPI value based on STR markers did not reach the threshold of 10 000 because there was a non-matching locus (D12S391). However, we confirmed the relationship between a mother and son for P5 using our microhaplotype combinations. Considering the good polymorphism and low mutation rates of our microhaplotype set, we believe that can be a complementary system for the routinely used STR markers. Given the high throughput of MPS, our panel can be combined with other microhaplotype panels such as Zhu's kinship analysis panel [48], to improve the forensic efficacy of paternity testing.

## 4. Conclusion

We developed a set of highly polymorphic microhaplotypes and evaluated their use for forensic analyses. The lengths of loci were limited to 200 bp and most amplicons were less than 300 bp, making them amenable to the MPS method. Moreover, several loci with small amplicons can be applied for the analysis of slightly degraded DNA samples. These markers will be particularly helpful for mixture analyses and for identifying individuals from East Asian populations. The population specificity of these markers will be helpful for inferring biogeographic ancestry. We believe that this microhaplotype set is a useful addition to forensic genetic testing.

Ethics. The ethics approval code: 2018-S194 and granted by the ethics committee of Central South University
Data accessibility. The datasets supporting this article have been uploaded as part of the electronic supplementary material.
Authors' contributions. A.K. and J.L. performed the experiments and wrote the manuscript, D.W. contributed to data interpretation and revised the whole manuscript, Z.Y. and S.S. helped with data acquisition and manuscript modification and L.Z. designed this research and modified the manuscript. All authors gave final approval for publication.
Competing interests. We have no competing interests.

Funding. This project was supported by the National Natural Science Foundation of China (NSFC, grant no. 81871533), the Natural Science Foundation of Hunan Province (grant no. 2017JJ3422) and the Shanghai Key Laboratory of Forensic Medicine Open Project (grant no. KF1815).

Acknowledgements. We thank the volunteers who contributed samples for this study.

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
