## [Reviewer comments · Royal Society Open Science]

Review History

RSOS-191937.R0 (Original submission)

Review form: Reviewer 1

Is the manuscript scientifically sound in its present form?

Yes

Are the interpretations and conclusions justified by the results?

Yes

Is the language acceptable?

No

Do you have any ethical concerns with this paper?

No

Have you any concerns about statistical analyses in this paper?

No

Recommendation?

Accept with minor revision (please list in comments)

Comments to the Author(s)

The authors present information on 20 microhaplotypes, each defined by 3 SNPs within a 200bp region with one SNP generally having three alleles detected. This marker panel is recommended for forensic uses. The work is straightforward and appears to be correct, although some minor changes are suggested.

The overall motivation for identifying this marker panel is not well described: what are the advantages over current STR typing panels? What are the advantages over sequencing the current STR markers? What are the plans to establish frequency databases for this panel?

The quantity A_e is mentioned several times but is not defined in this manuscript. What is it? How, exactly, does it predict mixture detection? How is match probability MP defined? How does MP differ from RMP?

The HWE and LD tests do not address multi-locus dependencies. How does actual match probability (not the product of single-locus probabilities) change with the number of loci? Is 20 loci sufficient for this system to be adopted by forensic scientists? If not, how many more are needed?

Why was F_{st} calculated? How is it to be used by forensic scientists?

It is not good style to present a number such as 67 following 17 '9's. Use accepted scientific notation.

Why was the non-matching STR locus for mother-son pairs ignored?

The manuscript has several lapses of correct English.

Review form: Reviewer 2

Is the manuscript scientifically sound in its present form?

Yes

Are the interpretations and conclusions justified by the results?

Yes

Is the language acceptable?

No

Do you have any ethical concerns with this paper?

No

Have you any concerns about statistical analyses in this paper?

Yes

Recommendation?

Major revision is needed (please make suggestions in comments)

Comments to the Author(s)

The authors have attempted to identify better microhaplotypes for individualization, paternity testing, and estimation of biogeographic ancestry. Tri-allelic SNPs have been the foci for the microhaplotypes they have identified. They have described the search criteria they used to

identify the SNPs constituting the microhaplotypes and have designed primers to amplify and sequence the loci.

They have presented many relevant calculations and illustrations, in two cases too much information:

(1) PIC is not a meaningful forensic statistic. It was defined in the 1980's as a statistic relevant to linkage analyses. It should be deleted from the study.

(2) Statistical tests of LD for loci on different chromosomes (Supplementary Table 2; Supplementary Figure 1) are simply measuring noise or the consequence of the structure of the population relationships. The calculation seems meaningless except for loci on the same chromosome. Moreover, it is quite disconcerting to see a seemingly random order of the loci. A genome order by chromosome number as in Table 1 would be simpler to follow but only the within-chromosome values are worth presenting.

There are a few places where the English should be improved because the meaning is not clear. In general the problems are not ones leading to confusion but a good proofreading is essential. The following are just a couple of points among the many instances of incorrect word choice or omitted words or number errors.

page 3, l. 5 number error "rate among SNPs are"

page 3, l. 8: the meaning is probably interpretable but the sentence is strictly jibberish

page 4, l. 59: an orphan "t"

page 4, l.49: "S14" is the name of a specific DNA sample in the second usage in the line. The absence of an article makes the first usage unclear and confusing. In all places it should be "...the S14 sample..."

l. 58 "length of 20 loci range" number error. Also "the length of 13 loci are"

page 5, l. 10: What is meant by "might be affected by sample content of unrelated individuals"?

Did the authors expect two different samples of Han Chinese to be identical? Given the variation within any population two samples are expected to differ by sampling error along.

ll. 16-17 This is not a correct definition of Ae .

ll. 19-22: There are many elements that make no sense including averages outside the ranges just stated. Scientific notation would be better than the string of 9's.

Figure 3 might make note of the PEL being the least admixed of the American samples and that it shows that known pattern with these markers. The European admixture is clear, as is some African admixture, in the other AMR samples.

Supplemental Table 4 provides the microhaplotype frequencies summarized from 1000 Genomes. It is possible to see that the triallelic SNPs have reasonable frequencies for all alleles (based on the few I checked) but it would be nice to have a separate table with just those 20 SNPs and their frequencies in all the populations. That would satisfy my curiosity since many SNPs that are listed as tri-allelic have the third allele at a near zero frequency because it was seen only a very few times in only one or a few populations. In this study it seems that the tri-allelic SNP is a significant contributor to the overall heterozygosity of the loci. That is a point worth emphasizing.

Supplemental Figure 2 shows that the 30 Han sequenced were quite polymorphic, validating the selection of the microhaplotype loci. However, the actual numbers should also be incorporated into Table 4 as an additional column with the five East Asian populations from 1000 Genomes.

Decision letter (RSOS-191937.R0)

17-Jan-2020

Dear Miss Kureshi,

The editors assigned to your paper ("Construction and forensic application of 20 highly polymorphic microhaplotypes") have now received comments from reviewers. The reviewers and Associate Editor consider the paper to be of significant interest and potentially meriting

publication. However, the reviewers raise a large number of points for clarification and revision, which will need to be addressed thoroughly before the paper can proceed. We would like you to revise your paper in accordance with the referee and Associate Editor suggestions which can be found below (not including confidential reports to the Editor). Please note this decision does not guarantee eventual acceptance.

Please submit a copy of your revised paper before 09-Feb-2020. Please note that the revision deadline will expire at 00.00am on this date. If we do not hear from you within this time then it will be assumed that the paper has been withdrawn. In exceptional circumstances, extensions may be possible if agreed with the Editorial Office in advance. We do not allow multiple rounds of revision so we urge you to make every effort to fully address all of the comments at this stage. If deemed necessary by the Editors, your manuscript will be sent back to one or more of the original reviewers for assessment. If the original reviewers are not available, we may invite new reviewers.

- Data accessibility

<http://datadryad.org/submit?journalID=RSOS&manu=RSOS-191937>

- Competing interests

- Authors' contributions

- Acknowledgements

- Funding statement

on behalf of Professor Peter Visscher (Associate Editor) and Steve Brown (Subject Editor)
openscience@royalsociety.org

Associate Editor's comments (Professor Peter Visscher):

Both expert referees were generally satisfied with the analyse/science, but have a long list of comments and suggestions, which all need to be addressed (thoroughly). In the light of these comments, the Editorial decision is somewhere in between 'minor' and 'major' revision.

The Editors will make a final decision once they have seen how the referee comments were dealt with.

Comments to Author:

Reviewers' Comments to Author:

Reviewer: 1

Comments to the Author(s)

The authors present information on 20 microhaplotypes, each defined by 3 SNPs within a 200bp region with one SNP generally having three alleles detected. This marker panel is recommended for forensic uses. The work is straightforward and appears to be correct, although some minor changes are suggested.

The overall motivation for identifying this marker panel is not well described: what are the advantages over current STR typing panels? What are the advantages over sequencing the current STR markers? What are the plans to establish frequency databases for this panel?

The quantity A_e is mentioned several times but is not defined in this manuscript. What is it? How, exactly, does it predict mixture detection? How is match probability MP defined? How does MP differ from RMP ?

The HWE and LD tests do not address multi-locus dependencies. How does actual match probability (not the product of single-locus probabilities) change with the number of loci? Is 20 loci sufficient for this system to be adopted by forensic scientists? If not, how many more are needed?

Why was F_{st} calculated? How is it to be used by forensic scientists?

It is not good style to present a number such as 67 following 17 '9's. Use accepted scientific notation.

Why was the non-matching STR locus for mother-son pairs ignored?

The manuscript has several lapses of correct English.

Reviewer: 2

Comments to the Author(s)

The authors have attempted to identify better microhaplotypes for individualization, paternity testing, and estimation of biogeographic ancestry. Tri-allelic SNPs have been the foci for the microhaplotypes they have identified. They have described the search criteria they used to identify the SNPs constituting the microhaplotypes and have designed primers to amplify and sequence the loci.

They have presented many relevant calculations and illustrations, in two cases too much information:

(1) PIC is not a meaningful forensic statistic. It was defined in the 1980's as a statistic relevant to linkage analyses. It should be deleted from the study.

(2) Statistical tests of LD for loci on different chromosomes (Supplementary Table 2; Supplementary Figure 1) are simply measuring noise or the consequence of the structure of the population relationships. The calculation seems meaningless except for loci on the same chromosome. Moreover, it is quite disconcerting to see a seemingly random order of the loci. A genome order by chromosome number as in Table 1 would be simpler to follow but only the within-chromosome values are worth presenting.

There are a few places where the English should be improved because the meaning is not clear. In general the problems are not ones leading to confusion but a good proofreading is essential. The following are just a couple of points among the many instances of incorrect word choice or omitted words or number errors.

page 3, l. 5 number error "rate among SNPs are"

page 3, l. 8: the meaning is probably interpretable but the sentence is strictly jibberish

page 4, l. 59: an orphan "t"

page 4, l.49: "S14" is the name of a specific DNA sample in the second usage in the line. The absence of an article makes the first usage unclear and confusing. In all places it should be "...the S14 sample..."

l. 58 "length of 20 loci range" number error. Also "the length of 13 loci are"

page 5, l. 10: What is meant by "might be affected by sample content of unrelated individuals"?

Did the authors expect two different samples of Han Chinese to be identical? Given the variation within any population two samples are expected to differ by sampling error along.

ll. 16-17 This is not a correct definition of A_e .

ll. 19-22: There are many elements that make no sense including averages outside the ranges just stated. Scientific notation would be better than the string of 9's.

Figure 3 might make note of the PEL being the least admixed of the American samples and that it shows that known pattern with these markers. The European admixture is clear, as is some African admixture, in the other AMR samples.

Supplemental Table 4 provides the microhaplotype frequencies summarized from 1000 Genomes. It is possible to see that the triallelic SNPs have reasonable frequencies for all alleles (based on the few I checked) but it would be nice to have a separate table with just those 20 SNPs and their frequencies in all the populations. That would satisfy my curiosity since many SNPs that are listed as tri-allelic have the third allele at a near zero frequency because it was seen only a very few times in only one or a few populations. In this study it seems that the tri-allelic SNP is a significant contributor to the overall heterozygosity of the loci. That is a point worth emphasizing.

Supplemental Figure 2 shows that the 30 Han sequenced were quite polymorphic, validating the selection of the microhaplotype loci. However, the actual numbers should also be incorporated into Table 4 as an additional column with the five East Asian populations from 1000 Genomes.

Author's Response to Decision Letter for (RSOS-191937.R0)

See Appendix A.

RSOS-191937.R1 (Revision)

Review form: Reviewer 1

Is the manuscript scientifically sound in its present form?

Yes

Are the interpretations and conclusions justified by the results?

Yes

Is the language acceptable?

Yes

Do you have any ethical concerns with this paper?

No

Have you any concerns about statistical analyses in this paper?

No

Recommendation?

Accept with minor revision (please list in comments)

Comments to the Author(s)

The authors/ changes have improved the manuscript.

There are still a few issues:

LD tests and r^2 statistics address only pairwise dependencies among marker frequencies, and do not address the validity of the product rule for multiple loci. The authors began to address this issue in Table S of their response, although the number of matches for 6 or more loci was too small to be informative. At face value, the authors demonstration that the actual matching proportion differs from the product rule prediction by a factor of 100 does not "support our previous conclusion about 20 microhaplotypes were independent."

The authors seem not to realize that F_{st} supplied the theta value needed for Balding-Nichols match probabilities.

The authors claim to calculate a "probability of detecting mixtures" but their explanation in the Results and Discussion section is not clear.

Review form: Reviewer 2

Is the manuscript scientifically sound in its present form?

Yes

Are the interpretations and conclusions justified by the results?

Yes

Is the language acceptable?

No

Do you have any ethical concerns with this paper?

No

Have you any concerns about statistical analyses in this paper?

Yes

Recommendation?

Accept with minor revision (please list in comments)

Comments to the Author(s)

The authors have responded well to the specific comments. However, the English is still not good especially in the Results and Discussion. But with a few exceptions the meaning will be clear enough. I would encourage the authors to have someone fluent in English to read the manuscript on the outlook for errors in number and in use of articles.

Page 2, line 27 "and Yoruba in Ibadan, Nigeria (YRI)." to be consistent

In Results on p.11 the two sentences in lines 22-24 are contradictory. This needs to be fixed.

line 32: "locus ability to deconvolute a DNA mixture"

line 43: "observed to have at least four alleles"

p.12, line 7: word choice. "separated from EUR" would be better

Decision letter (RSOS-191937.R1)

06-Mar-2020

Dear Miss Kureshi:

Manuscript ID RSOS-191937.R1 entitled "Construction and forensic application of 20 highly polymorphic microhaplotypes" which you submitted to Royal Society Open Science, has been reviewed. The comments of the reviewer(s) are included at the bottom of this letter.

Both reviewers are enthusiastic about the publication of your paper, but they raise a few final points that need to be dealt with and revised before we can accept the paper. We would also like you to reexamine the English across the manuscript and to do your best to improve this aspect of the paper (see the comments from the Associate Editor).

Please submit a copy of your revised paper before 29-Mar-2020. Please note that the revision deadline will expire at 00.00am on this date. If we do not hear from you within this time then it will be assumed that the paper has been withdrawn. In exceptional circumstances, extensions may be possible if agreed with the Editorial Office in advance. We do not allow multiple rounds of revision so we urge you to make every effort to fully address all of the comments at this stage. If deemed necessary by the Editors, your manuscript will be sent back to one or more of the original reviewers for assessment. If the original reviewers are not available we may invite new reviewers.

- Ethics statement

- Data accessibility

- Competing interests

- Authors' contributions

- Acknowledgements

- Funding statement

on behalf of Steve Brown (Subject Editor)
openscience@royalsociety.org

Associate Editor Comments to Author:

Comments to the Author:

The reviewers indicate the paper is improved but a number of matters remain to be resolved. Notably, clarity of writing and the quality of the written English - we certainly sympathise that it's a tricky language, but professional editing services are available to assist with this. You should seek advice from one or other of the services identified at <https://royalsociety.org/journals/authors/language-polishing/>, and you will be asked to provide evidence such as a certificate of English language editing on the submission of your revision. If you do not do so, and the Editors remain unsatisfied by your responses to the remaining changes to be made, we cannot guarantee the paper will be considered ready for publication. Good luck.

Reviewer comments to Author:

Reviewer: 2

Comments to the Author(s)

The authors have responded well to the specific comments. However, the English is still not good especially in the Results and Discussion. But with a few exceptions the meaning will be

clear enough. I would encourage the authors to have someone fluent in English to read the manuscript on the outlook for errors in number and in use of articles.

Page 2, line 27 "and Yoruba in Ibadan, Nigeria (YRI)." to be consistent

In Results on p.11 the two sentences in lines 22-24 are contradictory. This needs to be fixed.

line 32: "locus ability to deconvolute a DNA mixture"

line 43: "observed to have at least four alleles"

p.12, line 7: word choice. "separated from EUR" would be better

Reviewer: 1

Comments to the Author(s)

The authors/ changes have improved the manuscript.

There are still a few issues:

LD tests and r^2 statistics address only pairwise dependencies among marker frequencies, and do not address the validity of the product rule for multiple loci. The authors began to address this issue in Table S of their response, although the number of matches for 6 or more loci was too small to be informative. At face value, the authors demonstration that the actual matching proportion differs from the product rule prediction by a factor of 100 does not "support our previous conclusion about 20 microhaplotypes were independent."

The authors seem not to realize that F_{st} supplied the theta value needed for Balding-Nichols match probabilities.

The authors claim to calculate a "probability of detecting mixtures" but their explanation in the Results and Discussion section is not clear.

Author's Response to Decision Letter for (RSOS-191937.R1)

See Appendix B.

Decision letter (RSOS-191937.R2)

07-Apr-2020

Dear Miss Kureshi,

It is a pleasure to accept your manuscript entitled "Construction and forensic application of 20 highly polymorphic microhaplotypes" in its current form for publication in Royal Society Open Science.

Due to rapid publication and an extremely tight schedule, if comments are not received, your

paper may experience a delay in publication. Royal Society Open Science operates under a continuous publication model. Your article will be published straight into the next open issue and this will be the final version of the paper. As such, it can be cited immediately by other researchers. As the issue version of your paper will be the only version to be published I would advise you to check your proofs thoroughly as changes cannot be made once the paper is published.

on behalf of Prof Steve Brown (Subject Editor)
openscience@royalsociety.org

Appendix A

Dear Editors and Reviewers,

Thank you for your letter. I greatly appreciate both your help and that of the referees concerning improvement to this paper. We have studied their comments carefully and have made corrections which we hope meet their approval.

Reviewer: 1

Comments to the Author(s)

The authors present information on 20 microhaplotypes, each defined by 3 SNPs within a 200bp region with one SNP generally having three alleles detected. This marker panel is recommended for forensic uses. The work is straightforward and appears to be correct, although some minor changes are suggested.

1. The overall motivation for identifying this marker panel is not well described: what are the advantages over current STR typing panels? What are the advantages over sequencing the current STR markers? What are the plans to establish frequency databases for this panel?

Response: Thanks for the comment. We noticed that we overlooked the comparative overview between microhaplotype markers and conventional STR markers. In hence, we targeted on the advantages of microhaplotypes over STR markers and STR sequencing and made fully statement in the second paragraph of introduction. The statements are displayed as follows: At present, short tandem repeats (STRs) are the preferable markers in forensic genetics owing to the high polymorphism produced by its multi-allelic nature and generally applied for individual identification, biological relationship predication, and mixture analysis [14]. Capillary electrophoresis (CE) is the regular detecting method in forensic genetics for STR genotyping. STR has some shortcomings, such as high mutation rate, artificial peaks and small potential for ancestry identification [15][16]. STR mutation rate is 10^3 - 10^4 times that of SNP [17], which has the risk of causing false exclusion in paternity testing [18]. CE-based STRs usually generate artificial peaks like stutter peaks and -A peaks, these peaks affect analysis of unbalanced mixture samples [19]. MPS-based STR detection method addressed the artificial peaks generated by CE, but the read lengths of most MPS platform and the homopolymer sequencing error that generate during STR sequencing, as well as the complexity of data interpretation, limited the development of MPS for STRs [20][21][22]. Compared with CE-based STRs, the recommended MPS-based microhaplotypes are not affected by artificial peaks and are suitable for detecting minor donors in mixture deconvolution [23]. In addition, due to the rarely mutant SNPs component, microhaplotypes reduce the misinterpretations in paternity testing and provide biogeographical ancestry information [24][25]. Thereby, microhaplotypes could be great supplementary tools for STRs in forensic science.

The plans to establish frequency databases for this panel were added to the revised manuscript at the first part of results and discussion section as follow: We plan to improve frequency databases in the future so that this highly polymorphic panel could be applied to forensic caseworks. Samples of some ethnic groups will be collected with the help of other forensic laboratories to calculate allele frequencies and other forensic parameters.

2. The quantity A_e is mentioned several times but is not defined in this manuscript. What is it? How, exactly, does it predict mixture detection? How is match probability MP defined? How does MP differ from RMP?

Response: Thank you for your valuable advice about A_e . As Kidd described, A_e is a term “converts the unequal allele frequencies at a locus into a value that is equivalent to some number of equally frequent alleles” [1] and were calculated using the formula $1/\sum p_i^2$ (where p_i represents the frequency of allele i). The theoretical probability of detecting mixtures was increased with an increase of A_e so A_e was used for estimating the ability to qualitatively detect mixtures. For instance, assume two individual mixtures were detected using loci A (A_e value is 3) and B (A_e value is 4), respectively. The probability of detecting three alleles for loci A (P_A) is $(2p^2qr+2q^2pr+2r^2pq+4p^2qr+4q^2pr+4r^2pq) \times 2=0.4444$ while the probability of detecting three alleles for loci B (P_B) = 0.6563. The definition of A_e value was added at revised manuscript in sixth part (statistical analysis) of materials and methods as follows: Kidd defined the effective number of alleles (A_e) for a locus as the number of neutral alleles of equal frequency and can be calculated using the formula $1/\sum p_i^2$ (where p_i represents the frequency of allele i) [35].

We are sorry about our ambiguous description about the MP and RMP. Matching probability (MP) was defined as the probability that two randomly selected individuals have the same genotype at the tested locus [2,3]. $MP = \sum_{k=1}^m p_k^2$ (p_k is the frequency of k for each different genotype, and m is the number of different types.) The probability of discrimination (PD) is equal to $1 - MP$. The matching probability is the same concept of random matching probability (RMP). In order to avoid confusion of manuscript, we replaced “RMP” with “the combined matching probability”.

References

1. Kidd KK, Speed WC. 2015 Criteria for selecting microhaplotypes: Mixture detection and deconvolution. *Investig. Genet* (2015) 6:1.
2. Jones DA. 1972 Blood Samples: Probability of Discrimination. *J. Forensic Sci. Soc.* 12, 355–359.
3. Geroge FS. 1982 Biochemical markers of individuality. In *Forensic Science Handbook* (ed R.Saferstein), pp. 338–415.

3. The HWE and LD tests do not address multi-locus dependencies. How does actual match probability (not the product of single-locus probabilities) change with the number of loci? Is 20 loci sufficient for this system to be adopted by forensic scientists? If not, how many more are needed?

Response: Thanks for the comment. Multi-locus independencies were of great importance for a panel applied in forensic caseworks. The ideal panel is that all loci were on different chromosomes, but it is difficult to achieve. At present, HWE tests are used to test whether samples are randomly sampled and LD tests are commonly used methods for assessing multi-locus independencies in forensics [1], although these methods have their own shortcomings. In our study, we employed not only LD but also r^2 for further understanding of the linkage between multi-locus for the sake of caution. The results of these methods all suggested that 20 microhaplotypes were in a state of linkage disequilibrium.

Calculating the actual match probability is a very sensible and professional advice. We explore this question with 2554 genotype-known samples (2504 samples from 1000 genome project and 50 unrelated individual samples from our study). We used different numbers of loci (from NO.1 to NO.20 locus, following the genome order) to calculate the actual combined matching probability and the corresponding combined matching probability. Detailed results are shown in the table below (Table S). There are not any of two individuals have the same genotype when the number of loci is greater than 10.

When considering only 10 loci, the actual combined matching probability is 3.06×10^{-7} , and the combined matching probability (calculated through product of MP) is 3.6×10^{-9} . The comparisons further support our previous conclusion about 20 microhaplotypes were independent. The combined matching probability for 20 microhaplotypes is 2.37×10^{-18} , similar to 13 CODIS STRs (5.02×10^{-16}) and SNP for ID 52-plex (5.0×10^{-19}). Moreover, the cumulative power of exclusion value is $1 - 1.928 \times 10^{-7}$, much greater than the CPE recommended for paternity testing, which is over $1 - 10^{-4}$. Thus, we believed that those 20 loci sufficient for this system to be adopted by forensic scientists for individual identification and paternity tests.

Table S. the actual combined matching probability and the combined matching probability through product.

number of loci	11-20	10	8	6
matching pairs	0	1	2	11
the actual combined matching probability	–	3.06×10^{-7}	6.13×10^{-7}	3.37×10^{-6}
the combined matching probability through product	4.06×10^{-10} - 2.37×10^{-18}	3.6×10^{-9}	2.25×10^{-7}	1.5×10^{-5}

Reference

1. Butler JM. 2015 Statistical Interpretation Overview. In *Advanced Topics in Forensic DNA Typing: Interpretation*, 213–237.

4. Why was *F-st* calculated? How is it to be used by forensic scientists?

Response: When the DNA profile detected in the crime scene could not match profile from available DNA databases, forensic scientists intend to obtain further information such as possible ancestral origin or phenotypic feature from DNA sample. We rely on this task on ancestry informative markers to differ biogeographical ancestry. The *F-st* values can help forensic scientists to estimate the populations distinguishing ability of forensic markers [1,2], and regraded as criteria for selecting markers with populations distinguishing ability [3,4]. In hence, we computed the pairwise *F-st* values for 26 populations from 1000 genome project based on 20 microhaplotype loci to examine the similarity of populations.

References

1. Chen P, Zhu J, Pu Y et al. 2017 Microhaplotype identified and performed in genetic investigation using PCR-SSCP. *Forensic Sci. Int. Genet.*

2. Chen P, Zhu W, Tong F et al. 2019 Identifying novel microhaplotypes for ancestry inference. *Int. J. Legal Med.* 133, 983–988.

3. Jung JY, Kang PW, Kim E et al. 2019 Ancestry informative markers (AIMs) for Korean and other East Asian and South East Asian populations. *Int. J. Legal Med.* 133, 1711–1719.

4: Vongpaisarnsin K, Listman JB, Malison RT et al. 2015 Ancestry informative markers for distinguishing between Thai populations based on genome-wide association datasets. *Leg. Med.* 17, 245–250.

5. It is not good style to present a number such as 67 following 17 '9's. Use accepted scientific notation.

Response: We are sorry about our inappropriate description. The unaccepted number expressions were corrected into scientific notation according to reviewers' suggestions.

6. Why was the non-matching STR locus for mother-son pairs ignored?

Response: Thanks for the comment. Due to the relatively high mutation rates of STR loci, non-matching STR locus are common in paternity tests [1]. In the original manuscript, we calculated the Combined paternity index (CPI) value of P5 mother-son pairs where PI value for the non-matching STR locus D12S391 was computed based on the specific probability of a mutant allele transitioning from parent to child [2]. In this case, the mutation procedure was assumed as allele 19 mutated to allele 18 (one step mutation). Thus, $PI \text{ (for D12S391)} = \mu_{19 \rightarrow 18} / p_{18}$ (μ indicate the mutation rate of D12S391 locus, p_{18} is allele frequency of locus 18) $= 0.5 \times \mu \times (1/10)^{s-1} \times 0.5$ (s is mutation step) $= 0.0048$. We then computed and found that the CPI value of this pair is less than 10,000, which means we cannot identify them as biological mother-child relationships. When combined PI from our microhaplotypes, CPI is larger than 10,000 then paternity confirmed. In fact, as mentioned in the original manuscript, the genotype of 20 microhaplotype loci for all 12 duos are in accordance with Mendel's law of inheritance. This is a good example of the advantages of the microhaplotype panel we developed over STRs, so we changed the discussions at revised manuscript in three part (Determination of biological relationship) of Results and Discussions as follows: In consideration of the good polymorphism and low mutation rates, we believed that this microhaplotype set can be a complementary system for routinely used STR.

References

1. Butler JM. 2015 Relationship Testing: Kinship Statistics. In Advanced Topics in Forensic DNA Typing: Interpretation. 349–401.
2. Fimmers R, Henke L, Henke J, Baur MP. 1992 How to Deal with Mutations in DNA-Testing. In Advances in Forensic Haemogenetics (ed SPM Rittner C.).

7. The manuscript has several lapses of correct English.

Response: The amendment of English expression following the recommendations of reviewers. We carefully revised the manuscript to avoid similar problems.

Reviewer: 2

Comments to the Author(s)

The authors have attempted to identify better microhaplotypes for individualization, paternity testing, and estimation of biogeographic ancestry. Tri-allelic SNPs have been the foci for the microhaplotypes they have identified. They have described the search criteria they used to identify the SNPs constituting the microhaplotypes and have designed primers to amplify and sequence the loci.

1. They have presented many relevant calculations and illustrations, in two cases too much information:

(1) PIC is not a meaningful forensic statistic. It was defined in the 1980's as a statistic relevant to

linkage analyses. It should be deleted from the study.

(2) Statistical tests of LD for loci on different chromosomes (Supplementary Table 2; Supplementary Figure 1) are simply measuring noise or the consequence of the structure of the population relationships. The calculation seems meaningless except for loci on the same chromosome. Moreover, it is quite disconcerting to see a seemingly random order of the loci. A genome order by chromosome number as in Table 1 would be simpler to follow but only the within-chromosome values are worth presenting.

Response: Thanks for the valuable comment. We deleted PIC statistic in the manuscript and tables. The Supplementary Table 2(Sheet 2) and Supplementary Figure 1 about LD test results were modified as Reviewer 2's opinion, as well as the interpretation on results of LD test. The modified interpretation as follow: The LD p values of microhaplotype pairs at the same chromosome are not <0.05 and showed no significant deviation from the expectations. The r^2 value between loci at the same chromosome are all <0.04 and are depicted as a matrix diagram in Supplementary Figure 1.

2. There are a few places where the English should be improved because the meaning is not clear. In general, the problems are not ones leading to confusion but a good proofreading is essential. The following are just a couple of points among the many instances of incorrect word choice or omitted words or number errors.

page 3, l. 5 number error "rate among SNPs are"

page 3, l. 8: the meaning is probably interpretable but the sentence is strictly gibberish

page 4, l. 59: an orphan "t"

page 4, l.49: "S14" is the name of a specific DNA sample in the second usage in the line. The absence of an article makes the first usage unclear and confusing. In all places it should be "... the S14 sample ... "

l. 58 "length of 20 loci range" number error. Also "the length of 13 loci are"

page 5, l. 10: What is meant by "might be affected by sample content of unrelated individuals"? Did the authors expect two different samples of Han Chinese to be identical? Given the variation within any population two samples are expected to differ by sampling error along.

ll. 16-17 This is not a correct definition of A_e .

ll. 19-22: There are many elements that make no sense including averages outside the ranges just stated. Scientific notation would be better than the string of 9's.

Response: We are so sorry about all mistakes that we made in manuscript and we are grateful about all recommendations that helpful for manuscript revisions. 1. Number error "rate" was corrected as "rates"; 2 the sentence was revised to "A number of microhaplotypes have been proposed before [25][26][27][28], but many of them have limited polymorphism."; 3. orphan "t" was deleted; 4. The expression "S14 sample" changed into "the S14 sample"; 5. Number error "length" was corrected as "lengths"; 6. We aware of our misunderstanding of the relationship between forensic statistics and sample content, we deleted the sentences "We also found the A_e values of mh04zha001 among CHB and CHS were 3.17 and 3.13, respectively. It indicated that the A_e value of mh04zha001 might be affected by sample content of unrelated individuals"; 7. We modified the definition of A_e into "This value represents to the number of alleles with the equal frequency that produce the same heterozygosity as a plurality of alleles with different frequencies"; 8. The unaccepted number

expressions were corrected and wrong forensic statistics in result section were revised.

3. Figure 3 might make note of the PEL being the least admixed of the American samples and that it shows that known pattern with these markers. The European admixture is clear, as is some African admixture, in the other AMR samples.

Response: Thanks for the comment. We also confirmed this after studying Figure 3 carefully. We added the following figure caption of Figure 3: At K=5, the distribution consistent with geographical regions, while AMR cannot be totally detached from EUR. Among AMR populations, PEL samples are the least admixed, and other samples of American show European admixture and some of African admixture, which is consistent with the known pattern.

4. Supplemental Table 4 provides the microhaplotype frequencies summarized from 1000 Genomes. It is possible to see that the tri-allelic SNPs have reasonable frequencies for all alleles (based on the few I checked) but it would be nice to have a separate table with just those 20 SNPs and their frequencies in all the populations. That would satisfy my curiosity since many SNPs that are listed as tri-allelic have the third allele at a near zero frequency because it was seen only a very few times in only one or a few populations. In this study it seems that the tri-allelic SNP is a significant contributor to the overall heterozygosity of the loci. That is a point worth emphasizing.

Response: Thanks for the comment. Since SNPs was selected according to allele frequencies in Han Chinese in Beijing (CHB) and Southern Han Chinese (CHS) from the 1000 Genome Project, tri-allelic SNPs have reasonable frequencies for all alleles in these populations. In other populations especially in geographically distant groups tri-allelic SNPs may have the third allele at a near zero frequency. The allele frequencies of tri-allelic SNPs for each microhaplotype were illustrated in Supplementary Table 5. We emphasize the effect of tri-allelic SNPs in first part of “results and discussion” as follow: To a certain extent, we reckon those tri-allelic SNPs in microhaplotypes contribute a lot to the polymorphism of each locus in Chinese Han populations.

5. Supplemental Figure 2 shows that the 30 Han sequenced were quite polymorphic, validating the selection of the microhaplotype loci. However, the actual numbers should also be incorporated into Table 4 as an additional column with the five East Asian populations from 1000 Genomes.

Response: Thanks for the comment. “50 unrelated Han Chinese” column was added into EAS group of Supplementary Table 4.

Appendix B

Dear Editors and Reviewers,

Thank you for your letter and for the reviewers' comments on our manuscript. Those valuable comments are very helpful for revising our paper and guiding our researches. Professional editing services from recommended websites were employed to improve clarity of writing and the quality of the written English (the editorial certificate was submitted as a Supplementary file). We have learned those comments carefully and made corrections which we hope meet approval.

Reviewer: 1

Comments to the Author(s)

1. LD tests and r^2 statistics address only pairwise dependencies among marker frequencies, and do not address the validity of the product rule for multiple loci. The authors began to address this issue in Table S of their response, although the number of matches for 6 or more loci was too small to be informative. At face value, the authors demonstration that the actual matching proportion differs from the product rule prediction by a factor of 100 does not "support our previous conclusion about 20 microhaplotypes were independent."

Response: Thank you for your comment. We are inspired by your suggestion and aware of insufficient of our research. We noticed that microhaplotype markers were not completely independent due to presence of population substructure. In hence, we decided to made adjustments for matching probability calculation based on Balding& Nichols formula. The results are displayed as follow: “The combined matching probability of 26 populations were calculated following the method proposed by Balding et al.[47], and ranged from 9.57×10^{-4} (MSL population) to 1.04×10^{-12} (STU population). The combined Matching probability of CHB population for unrelated individuals was 8.73×10^{-11} , suggesting this set can be used independently for personal identification.”

2. The authors seem not to realize that F_{ST} supplied the theta value needed for Balding-Nichols match probabilities.

Response: Thank you for your comment. We realized that the theta value is same as F_{ST} when mating within subpopulations is random. Thus, F_{ST} statistics would be helpful in our Balding-Nichols match probabilities calculation.

3. The authors claim to calculate a "probability of detecting mixtures" but their explanation in the Results and Discussion section is not clear.

Response: We are sorry about our unambiguous statement about probability of detecting mixtures. We supplied the explanation as follow: “ A_e values correlate with the ability of microhaplotype loci to detect and deconvolute DNA mixtures [46]. For instance, When a microhaplotype locus with an A_e value of 3.0 is applied for detecting a mixture of two unrelated individuals, the probability of there being a third allele was 0.4444 under the simple HWE model [35]. Hence, the maximum probability of

detecting a mixture for this locus was 0.4444; for a locus with an A_e value of 4.0, the maximum probability would be 0.65625. We utilized the minimal integral value of A_e for our probability calculation. The cumulative probability of detecting a mixture with the set of microhaplotypes was equal to $1 - (1 - 0.4444)^{15} (1 - 0.65625)^4 = 0.999997927$. A_e values convey information on the polymorphisms of markers, and we assessed how these could be used to analyses DNA mixtures. The larger the A_e values reflect the better capacity of detecting mixed DNA samples.”

Reviewer: 2

Comments to the Author(s)

The authors have responded well to the specific comments. However, the English is still not good especially in the Results and Discussion. But with a few exceptions the meaning will be clear enough. I would encourage the authors to have someone fluent in English to read the manuscript on the outlook for errors in number an in use of articles.

1. Page 2, line 27 “and Yoruba in Ibadan, Nigeria (YRI).” to be consistent

In Results on p.11 the two sentences in lines 22-24 are contradictory. This needs to be fixed.

line 32: “locus ability to deconvolute a DNA mixture”

line 43: “observed to have at least four alleles”

p.12, line 7: word choice. “separated from EUR” would be better

Response: Thank you for your advice. We are sorry about English errors in manuscript.

(1) The Page 2, line 27 has been corrected as follow: “and Yoruba in Ibadan, Nigeria (YRI)”; (2) Two sentences in p.11 lines 22-24 are modified as follow: “The LD p values of microhaplotype markers on the same chromosome showed no significant deviation from expectations, suggesting that these sites were in linkage equilibrium. To further evaluate LD, we calculated another parameter, r^2 (Supplementary Figure 1). The r^2 value between marker pairs on the same chromosome were all under 0.04, supporting the previous conclusion of LD tests.”; (3) line 32 was revised to “ A_e values correlate with the ability of microhaplotype loci to detect and deconvolute DNA mixtures”; (4) line 43 changed into: “most microhaplotypes had at least four alleles”; (5) In p.12, line 7, the word “departed” is replaced by “separated” as suggested. In addition, the revised manuscript was together checked by professional editing services to correct errors.